# SynthRL: Scaling Visual Reasoning with Verifiable Data Synthesis

## Abstract

Vision-language models (VLMs) trained via reinforcement learning with verifiable reward (RLVR) have shown notable progress in scaling test-time compute effectively. In this work, we investigate how synthesized RL data can further improve RLVR. To this end, we propose **SynthRL**—a scalable and guaranteed pipeline for automatic data scaling in reasoning-oriented RL training. SynthRL comprises three key stages: (1) selecting seed questions with appropriate distribution, (2) augmenting them into more challenging variants while preserving the original answers, and (3) a guaranteed verification stage that ensures near-perfect correctness and difficulty enhancement. Our empirical experiments demonstrate SynthRL's scalability and effectiveness. When applied to the MMK12 dataset, SynthRL synthesizes over 3.3K additional verifiable, challenging questions from approximately 8K seed samples. Models trained with our synthesized data achieve consistent gains across five out-of-domain visual math reasoning benchmarks, with a significant improvement over baseline models trained on seed data alone. Notably, detailed analysis reveals that the gains are more pronounced on the most challenging evaluation samples, highlighting SynthRL's effectiveness in eliciting deeper and more complex reasoning patterns.

## 1 Introduction

Reinforcement Learning with Verifiable Rewards (RLVR) has recently emerged as a promising paradigm, significantly enhancing the reasoning capabilities of language and vision-language models (Guo et al., 2025; Shao et al., 2024; Liu et al., 2025b; Yu et al., 2025; Yuan et al., 2025; Zeng et al., 2025). At the same time, the data-centric approaches are increasingly recognized as critical for advancing the boundary of model intelligence (Bai et al., 2025; Abdin et al., 2025; Luo et al., 2024; Bai et al., 2024; Xu et al., 2024; 2023). Motivated by these insights, we raise a critical yet underexplored challenge: *Can we scale the RLVR training data with correctness and distribution guarantees to achieve better performance?*

Directly addressing this challenge remains non-trivial, as it is difficult to formulate it as a standard optimization problem. Although existing data selection methods may offer partial solutions in terms of distribution (Zhou et al., 2023; Li et al., 2025b; Xia et al., 2024; Wettig et al., 2024; Liu et al., 2023b; Tong et al., 2024), they are constrained by the original data volume and distribution, being less effective in scenarios where data is originally scarce and biased (Guo et al., 2024; Li et al., 2025a; Dong et al., 2023). Instead, we pursue a complementary and more practical direction—data synthesis—guided by the intuition that **under RLVR settings, more challenging yet still correct training samples can provide richer learning signals**. To this end, we introduce **SynthRL**, a streamlined and scalable pipeline specifically designed to effectively scale the RLVR training data for VLMs.

Specifically, our synthesis strategy employs a straightforward generation process coupled with guaranteed verification—an approach tailored for reinforcement learning where answer verifiability is paramount. This automated yet effective pipeline operates via a three-stage process:

1. **Seed Data Selection**: Seed questions for synthesis are identified by analyzing the pass count of Monte Carlo rollout by the target model. Questions exhibiting high pass rates are selected, as their limited challenge to the target model offers minimal training signals, rendering them ideal for complexity enhancement.

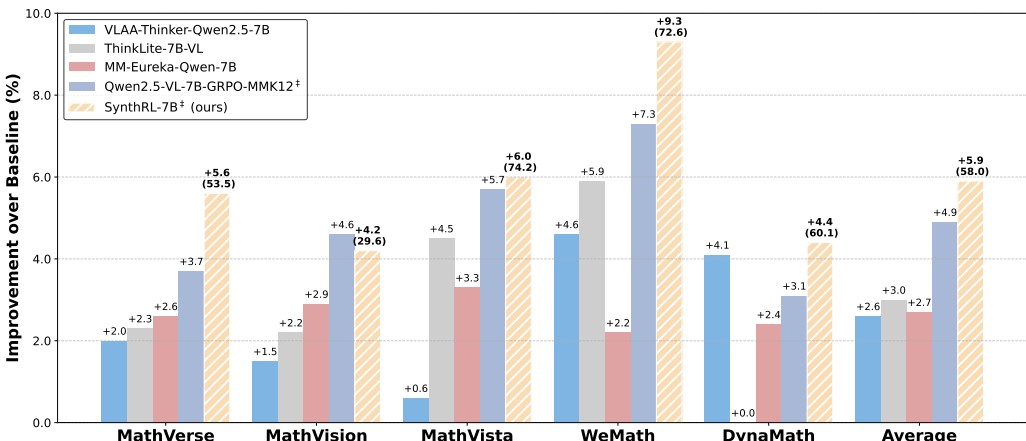

Figure 1: Improvement over baseline Qwen2.5-VL-7B-Instruct on five out-of-domain visual mathematical reasoning benchmarks: MathVerse, MathVision, MathVista, WeMath, and DynaMath. The chart compares performance of five different models across these benchmarks. The ‡ symbol indicates models trained by ourselves, which includes both Qwen2.5-VL-7B-GRPO-MMK12‡ and SynthRL-7B‡ (ours). SynthRL-7B additionally uses synthesized samples. The exact accuracy percentages for SynthRL-7B are shown in parentheses above each bar.

2. **Targeted Synthesis**: A powerful VLM is leveraged to generate more challenging variants of the selected questions while preserving the original ground-truth answers. This is achieved using minimal prompting that prioritizes an escalation in difficulty by requiring deeper reasoning.

3. **Verification**: A guaranteed verification step to filter synthesized data, confirming question validity, answer preservation, and an actual increase in difficulty. With the propose-solve mechanism, this verification ensures near-perfect correctness of newly synthesized training samples.

This pipeline efficiently scales existing datasets with more valuable training examples without human intervention. Applied to the MMK12 (Meng et al., 2025) dataset, our method generated over 3.3k verified harder questions from approximately 8k seed samples. Models trained with our synthesized data demonstrated substantial improvements across five out-of-domain visual math reasoning benchmarks (MathVerse (Lu et al., 2023), MathVision (Wang et al., 2024a), MathVista (Lu et al., 2023), WeMath (Qiao et al., 2024), and DynaMath (Zou et al., 2024)). For instance, significant performance gains were observed compared to models trained on seed data alone, including boosts of +1.9% on MathVerse, +2.0% on WeMath, and +1.3% on DynaMath using the 8k seed dataset. Notably, this positive impact on performance is consistently observed across various data scales. Detailed analysis reveals these improvements are most pronounced on challenging evaluation examples, confirming our approach's effectiveness in addressing complex reasoning scenarios.

## 2 RELATED WORKS

**Vision-language model reasoning.** Vision-Language Models (VLMs) have rapidly evolved from foundational integration techniques (Alayrac et al., 2022; Li et al., 2023b) and effective visual instruction tuning (Liu et al., 2023a; 2024; Li et al., 2024b;a) to specialized mathematical reasoning approaches like Math-LLaVA (Shi et al., 2024) and MAVIS (Zhang et al., 2024b). While advanced models like GPT-4o (Hurst et al., 2024) and Gemini (Gemini Team, 2023) show strong general visual understanding, a gap persists in robust visual reasoning requiring sophisticated analysis and complex inference. Reinforcement Learning (RL) is emerging to address this, extending from methods enhancing LLM reasoning (Guo et al., 2025; Shao et al., 2024; Kimi Team, 2025a). For VLMs, R1-type RL applications have shown success in specific subdomains like geometry and object counting (Peng et al., 2025; Huang et al., 2025; Chen et al., 2025b; Deng et al., 2025). Notably, recent studies (Meng et al., 2025; Yang et al., 2025; Liu et al., 2025a) has applied rule-based RL to achieve significant gains in broader multimodal mathematical reasoning for VLMs without in-domain training data.

**Data synthesis.** Data synthesis is vital for VLMs, providing scalable, diverse, and high-quality training data to enhance performance across applications (Cui et al., 2024; Wang et al., 2024b; Li

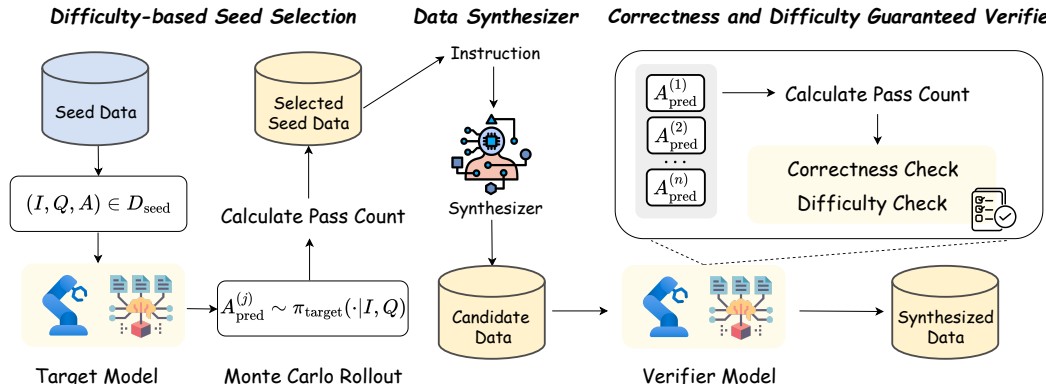

Figure 2: Illustration of our **SynthRL** pipeline. (1) **Difficulty-based Seed Selection** identifies suitable questions based on Monte Carlo rollout pass rates, (2) **Data Synthesizer** transforms selected questions into more challenging variants while preserving the original answer $A$, and (3) **Correctness and Difficulty Guaranteed Verifier** ensures both answer preservation and increased difficulty.

et al., 2023a). Initially focused on improving instruction following capabilities (Liu et al., 2023a; 2024) and aligning with human preferences through methods like multi-turn conversations and feedback mechanisms (Li et al., 2024d;c), recent research increasingly employs data synthesis to advance visual reasoning Zhang et al. (2024b); Yao et al. (2024); Luo et al. (2025). This newer focus includes generating sophisticated datasets for complex instructions or using techniques such as reverse chain-of-thought (Zhou et al., 2025; Du et al., 2025; Hu et al., 2025) to address tasks in geometric (Deng et al., 2024), mathematical (Shi et al., 2024), and navigational reasoning (Zhou et al., 2024), thereby significantly expanding VLM reasoning capabilities. However, leveraging data synthesis for RL training in VLMs remains a largely underexplored frontier.

## 3 SYNTHRL: SCALABLE AND VERIFIABLE DATA SYNTHESIS

We propose an automated and guaranteed pipeline for synthesizing more challenging RL training data, as illustrated in Figure 2. Our approach (1) refines the seed task distribution through difficulty assessment (Section 3.2), (2) employs a synthesizer to generate harder variants of these questions (Section 3.3), and (3) validates these variants with exact correctness and difficulty guarantees (Section 3.4). This methodology unlocks another smart way of data synthesis for reasoning-oriented RL, where a more challenging data distribution and strict answer correctness are crucial. The detailed algorithmic procedure of our approach is provided in Appendix H.

### 3.1 PRELIMINARY: REINFORCEMENT LEARNING WITH VERIFIABLE REWARDS

Before presenting our pipeline, we briefly outline the Reinforcement Learning with Verifiable Rewards (RLVR) framework. RLVR requires only a dataset $\mathcal{D} = \{(x, y^*)\}$ of inputs and correct outputs, without annotated reasoning steps. The model generates its own reasoning steps and receives a verifiable reward $r(y, y^*)$ based on the final answer. The policy $\pi_\theta$ is trained to maximize the expected reward: [1]

$$\mathcal{J}_{\text{RLVR}}(\theta) = \mathbb{E}_{(x,y^*) \sim \mathcal{D}, y \sim \pi_\theta(\cdot|x)}[r(y, y^*)]. \tag{1}$$

A key challenge in RLVR is scalability, due to the high cost of annotated data. Our method, SynthRL, addresses this by synthesizing additional training examples to augment the dataset, enabling the model to learn from both curated and synthetic data.

### 3.2 DIFFICULTY-BASED SEED SELECTION

**Difficulty assessment.** The first step in our synthesis pipeline is selecting suitable questions from a seed dataset $\mathcal{D}_{\text{seed}}$. Suitability is based on the question's difficulty relative to a specific VLM, the *target model* $\pi_{\text{target}}$. This model serves both as the initial policy for RL training and as the benchmark

---
[1]We implement this using Group Relative Policy Optimization (GRPO), detailed in Appendix B.

for assessing question difficulty. We treat difficulty as model-dependent, recognizing that a question may be easy for one model but hard for another. To assess question difficulty for $\pi_{\text{target}}$, we apply a Monte Carlo rollout procedure. For each image-question-answer triplet $(I, Q, A) \in \mathcal{D}_{\text{seed}}$, we define the *rollout pass count* as:

$$C_{\text{pass}}(I, Q, A; \pi_{\text{target}}) = \sum_{j=1}^{N} \mathbb{I}(A_{\text{pred}}^{(j)} = A) \tag{2}$$

where $A_{\text{pred}}^{(j)}$ is the answer predicted by $\pi_{\text{target}}$ for $(I, Q)$ in the $j$-th stochastic forward pass, sampled as $A_{\text{pred}}^{(j)} \sim \pi_{\text{target}}(\cdot|I, Q)$; $N$ is the number of Monte Carlo rollouts ($N = 16$ in our context by default); and $\mathbb{I}(\cdot)$ is the indicator function, returning 1 if its argument is true, and 0 otherwise. $C_{\text{pass}}$ ranges from 0 to $N$, with lower values indicating harder questions for $\pi_{\text{target}}$, as the model less consistently predicts the correct answer. Evaluating $C_{\text{pass}}$ across $\mathcal{D}_{\text{seed}}$ helps identify questions that are too easy (i.e., high $C_{\text{pass}}$), which can then be targeted for transformation into more challenging variants. Selection criteria (e.g., thresholds) can be tuned based on downstream task requirements.

**Difficulty-aware selection.** For each question-answer pair $(I, Q_{\text{ori}}, A)$ in the processed dataset $\mathcal{D}_{\text{seed}}$, we compute its rollout pass count $c_{\text{ori}} = C_{\text{pass}}(I, Q_{\text{ori}}, A; \pi_{\text{target}})$ using Equation 2 with respect to the target model $\pi_{\text{target}}$. As shown in Figure 3, these counts are heavily skewed toward the extremes, with many samples either consistently failed ($c_{\text{ori}} \approx 0$) or solved ($c_{\text{ori}} \approx N$). Since such extremes offer limited gradient signals for RL training (Yu et al., 2025; Yuan et al., 2025), we focus on questions the model solves reliably, selecting those with $c_{\text{ori}} \geq 12$ as inputs for the synthesis stage (Section 3.3).

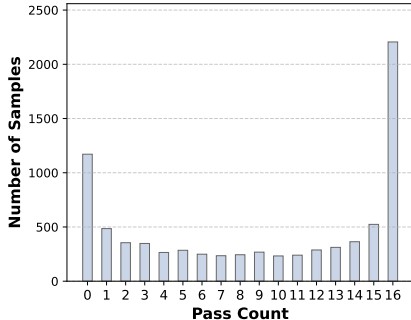

Figure 3: Distribution of rollout pass count on MMK12.

### 3.3 DATA SYNTHESIZER

The Synthesizer module generates more challenging variants of selected questions while preserving the original ground truth answer. For each sample $(I, Q_{\text{ori}}, A)$ from $\mathcal{D}_{\text{seed}}$, selected for its high rollout pass count (Section 3.2), a powerful general-purpose VLM ($\phi$) transforms $Q_{\text{ori}}$ into a candidate question requiring deeper reasoning.

For every input sample $(I, Q_{\text{ori}}, A)$, the synthesizer aims to produce a candidate question. The synthesis VLM is prompted with only the image $I$ and the original question $Q_{\text{ori}}$. The specific prompt template used is:

---

**Synthesizer Prompt**

Given an image and the following question, transform it into a significantly more challenging version that requires deeper reasoning but maintains the same answer.

Original Question:
{question}
Your Response Format:
New Question: {Your transformed question}

---

In this stage, the placeholder "{question}" is replaced with $Q_{\text{ori}}$, while the ground truth answer $A$ is deliberately withheld from the synthesis VLM. This setup compels the model to focus on the semantic relationship between $Q_{\text{ori}}$ and the image $I$, rather than relying on $A$ to produce superficial paraphrases. Consequently, it fosters the generation of questions that require deeper visual reasoning yet remain answerable with $A$. The output for each input $(I, Q_{\text{ori}}, A)$ is a candidate triplet $(I, Q_{\text{cand}}, A)$, where $Q_{\text{cand}}$ is a synthesized variant of $Q_{\text{ori}}$, later evaluated by the verifier module (Section 3.4) for quality and difficulty.

### 3.4 CORRECTNESS AND DIFFICULTY GUARANTEED VERIFIER

The verifier module validates synthesized questions, ensuring both task validity and difficulty increase.

**Candidate Evaluation.** For each candidate question $Q_{\text{cand}}$ generated from an original sample with rollout pass count $c_{\text{ori}}$, we apply the same rollout pass count metric as in Equation 2:

$$c_{\text{cand}} = C_{\text{pass}}(I, Q_{\text{cand}}, A; \pi_{\text{verifier}}) \tag{3}$$

**Verification Criteria.** A candidate question is deemed valid if it meets both of the following conditions:

1. **Correctness Criterion:** $c_{\text{cand}} \geq T_{\text{min}}$, ensuring the question remains answerable with the original answer. Here, $T_{\text{min}}$ represents the minimum number of successful rollouts required to consider a question correct. When a candidate question passes this threshold, it provides strong evidence that the question is valid and correctly preserves the original answer.
2. **Difficulty Criterion:** $c_{\text{cand}} \leq c_{\text{ori}} - \Delta_{\text{hard}}$, confirming the candidate question is measurably more difficult than the original. The parameter $\Delta_{\text{hard}}$ defines the minimum required increase in difficulty, measured as a reduction in pass count.

**Achieving Guaranteed Synthesis.** Our verification guarantees stem from a key design choice: the synthesizer is instructed to create harder questions with the same answer. Though the synthesizer aims to preserve the answer, not every generated question will succeed. The verifier resolves this uncertainty by evaluating each candidate against the original answer using the target model. When $\pi_{\text{target}}$ reaches the original answer a reasonable number of times (meeting the Correctness Criterion), it confirms the question is both valid and preserves the intended answer. Simultaneously, the Difficulty Criterion ensures only questions that genuinely challenge the model are accepted.

The final output of our three-stage pipeline is a collection of verified triplets $(I, Q_{\text{cand}}, A)$, each representing a harder variant of an original question designed to provide more informative gradient training for reinforcement learning fine-tuning.

## 4 DATASET

### 4.1 SEED AND SYNTHESIZED DATASETS

**Seed Dataset.** We use MMK12 (Meng et al., 2025) as our seed dataset, consisting of 8,099 question-answer pairs. For reliable verification in our pipeline, we preprocess the dataset by converting multiple-choice questions to free-form answer format and removing Yes/No questions. This preprocessing prevents reward hacking through random guessing during the verification stage, resulting in our seed dataset with 8,072 open-ended answers. For data scaling effect analysis, we also create 2k and 4k versions of the seed dataset as detailed in the Appendix D.

**Synthesized Dataset.** We use `Gemini-2.5-Flash-Preview-04-17` (Gemini Team, 2023) as our synthesizer model $\phi$. We select source questions with high rollout pass counts (at least 12 out of 16 successful predictions) from $\mathcal{D}_{\text{seed}}$ for transformation. For verification, we set the solvability criterion threshold $T_{\text{min}} = 4$ to guarantee question validity and answer preservation, and the difficulty criterion $\Delta_{\text{hard}} = 2$ to ensure candidates are measurably more challenging than their original versions. This process yields 3,380 verified harder variants, each preserving the original ground truth answer. We refer to the combined dataset of original MMK12 questions and their synthesized variants as $\mathcal{A}$-MMK12, totaling 11,452 samples. We apply the same synthesis process to the 2k and 4k versions. examples of our synthesized questions are provided in Appendix I.

### 4.2 DATA ANALYSIS

To understand our synthesized dataset's characteristics, we analyze pass rate distributions and reasoning complexity. The left side of Figure 4 compares the original MMK12 dataset with our complete $\mathcal{A}$-MMK12 dataset. The original MMK12 has a mean pass rate of 9.04, while $\mathcal{A}$-MMK12 shows a lower mean of 8.24, indicating increased overall difficulty.

The right side of Figure 4 provides a more focused comparison between the selected seed examples and their synthesized variants. Selected seed questions have a high mean pass rate of 15.10, while synthesized questions have a significantly lower mean of 6.33. This confirms our approach successfully creates more challenging variants from relatively easy seed examples.

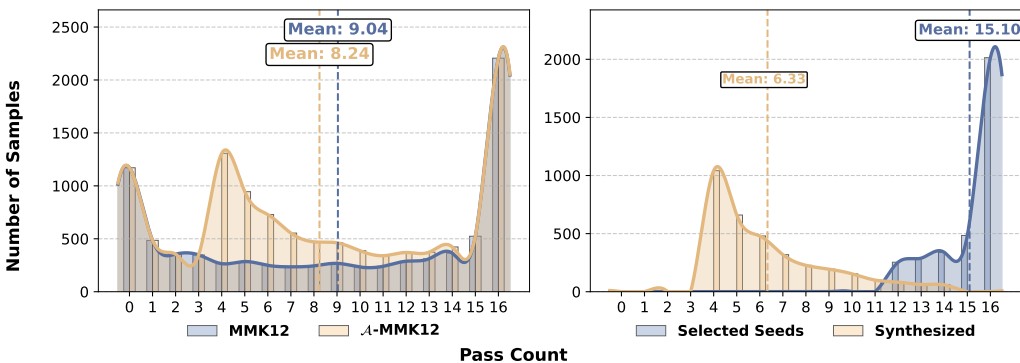

Figure 4: Pass rate distributions across datasets. The left figure compares the original MMK12 dataset with our complete $\mathcal{A}$-MMK12 dataset. The right figure compares the selected seed examples with their synthesized variants. The synthesized questions show a more balanced distribution across moderate difficulty levels, while seed questions cluster at the extremes.

The most notable difference appears in the distribution shape. The seed dataset shows high concentrations at the extreme ends of 0 and 16 passes, while synthesized questions display a more balanced distribution across intermediate difficulty levels from 4 to 14. This broader distribution provides a smoother difficulty progression during training, helping models develop better reasoning capabilities.

As shown in Figure 5, synthesized questions require more reasoning steps with a mean of 34.90 compared to original seed questions with a mean of 26.16. This 33% increase in reasoning steps indicates that our synthesis process creates problems requiring more elaborate reasoning chains. Questions with multi-step reasoning better exercise a model's ability to decompose problems and maintain coherent reasoning, essential for robust visual reasoning capabilities.

## 5 EXPERIMENTS

### 5.1 SETUP

**Implementation Details.** Following (Meng et al., 2025; Huang et al., 2025; Wang et al., 2025), we initialize our policy model with Qwen2.5-VL-7B-Instruct (Bai et al., 2025), well-suited for subsequent RL training due to its robust foundational capabilities. This same model serves as both the target model and verifier model in our methodology. For reinforcement learning training, we use the EasyR1 (Zheng et al., 2025) framework built on verl (Sheng et al., 2024), with specialized support for VLMs. All experiments are conducted using 8 NVIDIA H100 80GB HBM3

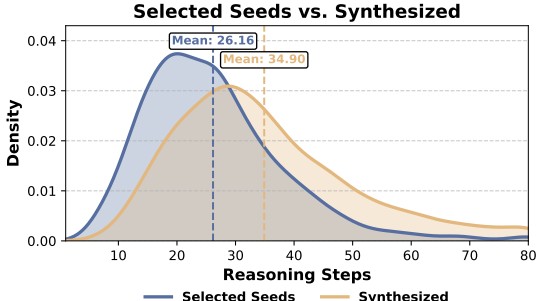

Figure 5: Distribution of reasoning steps between selected seed questions and synthesized questions.

GPUs with a global batch size of 128, a rollout batch size of 512, a rollout temperature of 1.0, a consistent learning rate of 1e-6, and 8 rollouts. We use EasyR1's standard reasoning template for training (see Appendix F). We train every dataset with sufficient training steps until convergence. Complete implementation details are provided in Appendix G.

Following recent research findings (Liu et al., 2025b; Yu et al., 2025), we remove the KL divergence constraint with the reference model in the GRPO algorithm to promote broader exploration. All parts of the model, including the vision encoder, are unlocked during training to maximize performance on visual reasoning tasks. Our main experiments compare two configurations: (1) *Baseline* models trained only on the original seed dataset, and (2) *SynthRL* models trained on $\mathcal{A}$-MMK12.

**Evaluation Benchmarks.** To assess model performance, we implement a comprehensive evaluation strategy across multiple benchmarks. We examine out-of-domain generalization capabilities using

Table 1: Performance comparison across visual reasoning benchmarks. Accuracy scores (%) are reported for each benchmark. **Bold values** indicate best performance, underlined values indicate second best. Models marked with $\star$ are evaluated using our evaluation pipeline. Dataset sizes are color-coded: SFT data, RL data, and synthesized RL data.

| Model | #Data | Benchmark Accuracy (%) | | | | | |
|---|---|---|---|---|---|---|---|
| | | MathVerse | MathVision | MathVista | WeMath | DynaMath | Avg. |
| *Close-source Models* | | | | | | | |
| Claude3.7-Sonnet (Anthropic, 2025) | – | 52.0 | 41.3 | 66.8 | – | – | – |
| GPT-4o (Hurst et al., 2024) | – | 50.2 | 30.4 | 63.8 | – | – | – |
| Gemini2.0-flash-001 (Gemini Team, 2023) | – | 59.3 | 41.3 | 70.4 | – | – | – |
| *Open-source Models* | | | | | | | |
| LLaVA-OneVision-7B (Li et al., 2024b) | – | 26.2 | – | 63.2 | – | – | – |
| Kimi-VL-16B (Kimi Team, 2025b) | – | 44.9 | 21.4 | 68.7 | – | – | – |
| Mulberry-7B (Yao et al., 2024) | – | – | – | 63.1 | – | – | – |
| InternVL-2.5-8B-Instruct (Chen et al., 2024) | – | 39.5 | 19.7 | 64.4 | – | – | – |
| Qwen-2.5-VL-7B-Instruct (Bai et al., 2025) | – | 47.9 | 25.4 | 68.2 | 63.3 | 55.7 | 52.1 |
| *RL-tuned Models with Verifiable Reward* | | | | | | | |
| R1-VL-7B (Zhang et al., 2025) | 260K+10K | 40.0 | 24.7 | 63.5 | – | – | – |
| Vision-R1-7B (Huang et al., 2025) | 200K+10K | 52.4 | – | 73.5 | – | – | – |
| R1-OneVision-7B$^\star$ (Yang et al., 2025) | 155K+10K | 46.1 | 22.5 | 63.9 | 62.1 | 53.7 | 49.7 |
| OpenVLThinker-7B$^\star$ (Deng et al., 2025) | 35K+15K | 48.0 | 25.0 | 71.5 | 67.8 | 57.5 | 54.0 |
| MM-Eureka-Qwen-7B$^\star$ (Meng et al., 2025) | 15K | 50.5 | 28.3 | 71.5 | 65.5 | 58.1 | 54.8 |
| ThinkLite-7B-VL$^\star$ (Wang et al., 2025) | 11K | 50.2 | 27.6 | 72.7 | 69.2 | 55.7 | 55.1 |
| VLAA-Thinker-Qwen2.5-7B$^\star$ (Chen et al., 2025a) | 126K+25K | 49.9 | 26.9 | 68.8 | 67.9 | 59.8 | 54.7 |
| *SynthRL$^\star$ (Ours)* | | | | | | | |
| *2K* | | | | | | | |
| MMK12 | 2K | 51.1 | 28.2 | 70.7 | 70.2 | 58.2 | 55.8 |
| $\mathcal{A}$-MMK12 | 2K+0.8K | 50.5 | 29.7 | 72.4 | 68.7 | 59.0 | 56.0 |
| *4K* | | | | | | | |
| MMK12 | 4K | 50.3 | 29.8 | 73.7 | 70.1 | 58.9 | 56.5 |
| $\mathcal{A}$-MMK12 | 4K+1.6K | 52.5 | 29.0 | 74.0 | 70.5 | 60.0 | 57.2 |
| *8K* | | | | | | | |
| MMK12 | 8K | 51.6 | **30.0** | 73.9 | 70.6 | 58.8 | 57.0 |
| $\mathcal{A}$-MMK12 | 8K+3.3K | **53.5** | 29.6 | **74.2** | **72.6** | **60.1** | **58.0** |

five specialized visual reasoning datasets: MathVerse (Zhang et al., 2024a), MathVision (Wang et al., 2024a), MathVista (Lu et al., 2023), WeMath (Qiao et al., 2024) and DynaMath (Zou et al., 2024).

For consistent evaluation across models, we develop a standardized evaluation suite capable of assessing both our trained checkpoints and most publicly available R1-related checkpoints. We use vLLM (Kwon et al., 2023) for efficient inference acceleration (denoted with $\star$), while incorporating reported results for models where direct evaluation was not feasible. Response evaluation uses greedy decoding with Gemini-2.0-Flash-001 (Gemini Team, 2023) as the judge for parsing generated outputs. We follow each model's provided system prompts and output formatting rules, though small differences from published results may exist due to our specific judge model and evaluation setup. Following the setting from (Zeng et al., 2025), we report the performance of the checkpoint that obtains the best average performance on the 5 benchmarks for all experiments.

## 5.2 RESULTS

**Main Finding 1: Out-of-domain generalization.** Our primary experiments in Table 1 show that SynthRL consistently improves performance across multiple out-of-domain visual reasoning benchmarks. At the 8K data scale, the model trained with the $\mathcal{A}$-MMK12 dataset achieves 58.0% average accuracy compared to 57.0% for the baseline model trained only on the seed MMK12 dataset. We observe significant improvements across individual benchmarks, with MathVerse accuracy increasing from 51.6% to 53.5% and WeMath from 70.6% to 72.6%. These results demonstrate that our synthetic data enhances generalization to unseen problem distributions.

**Main Finding 2: Data scaling effect.** The performance gap between $\mathcal{A}$-MMK12 and MMK12 is modest at the 2K scale (56.0% vs 55.8%), but widens considerably as more seed data becomes available, reaching +0.7% with 4K and +1.0% with 8K seed examples. This pattern suggests our synthesis

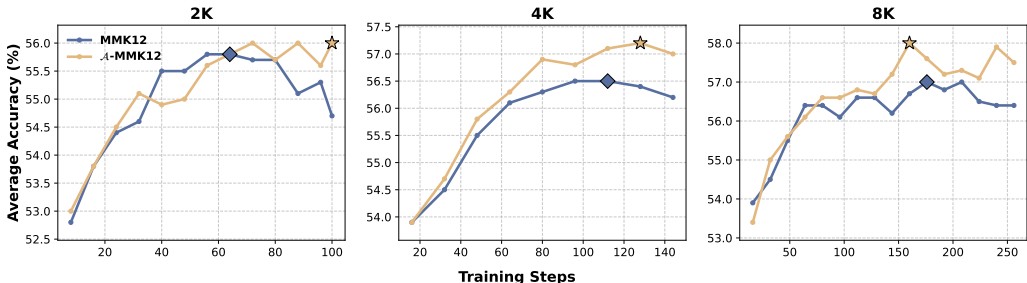

Figure 6: Performance on evaluation benchmarks across training steps for models trained on seed data (MMK12) versus synthesize-augmented data ($\mathcal{A}$-MMK12) at different data scales (2K, 4K, and 8K). Peak performance for $\mathcal{A}$-MMK12 and MMK12 are indicated by stars and markers, respectively.

Table 2: Average accuracy (%) by difficulty level across all five benchmarks.

| Method | 2K | | | 4K | | | 8K | | |
|---|---|---|---|---|---|---|---|---|---|
| | Easy | Med. | Hard | Easy | Med. | Hard | Easy | Med. | Hard |
| MMK12 | 67.2 | 54.3 | 44.9 | 67.9 | 55.4 | 46.4 | 69.3 | 54.9 | 46.8 |
| $\mathcal{A}$-MMK12 | 67.0 | 54.6 | 45.5 | 67.8 | 56.0 | 48.1 | 68.8 | 56.6 | 48.4 |
| $\Delta$ | -0.2 | -0.3 | +0.6 | -0.1 | +0.6 | +1.7 | -0.5 | +1.7 | +1.6 |

approach becomes more effective with larger, more diverse seed pools. Additionally, Figure 6 reveals that while both datasets lead to similar learning patterns initially, models trained on $\mathcal{A}$-MMK12 achieve higher peak performance across all data scales. Together, these results demonstrate that the benefits of our synthetic data augmentation become more pronounced with larger training datasets.

These findings demonstrate that our synthesis method complements traditional data scaling approaches, offering additional gains beyond what can be achieved through simply increasing the volume of original data. SynthRL's targeted generation of challenging variants creates a more effective training distribution for developing robust visual reasoning capabilities.

### 5.3 DIFFICULTY-BASED PERFORMANCE ANALYSIS

To precisely measure where our method provides the most value, we establish objective difficulty rankings for evaluation examples using the Bradley-Terry model and Elo rating system, similar to the approach used in Chatbot Arena (Chiang et al., 2024) for ranking large language models. We conduct pairwise comparisons of image-question pairs, with `Gemini-2.0-Flash-001` providing difficulty judgments across 128 battles per pair. This bootstrapped Elo-based methodology yields statistically robust difficulty scores that enable us to partition each benchmark dataset into three difficulty tiers: easy, medium, and hard.

Table 2 presents the average performance across all five benchmarks, grouped by difficulty level. Our analysis reveals that $\mathcal{A}$-MMK12 yields the largest improvements on the medium and hard subsets of examples. For the full 8K dataset, while $\mathcal{A}$-MMK12 performs slightly lower on easy examples (-0.5%), it shows clear gains on medium (+1.7%) and hard (+1.6%) examples. This pattern is consistent across data scales, where $\mathcal{A}$-MMK12 demonstrates its strongest advantage on the challenging problems.

These results demonstrate that our synthesis approach successfully targets complex reasoning challenges that are not adequately addressed by training on seed data alone. The performance shift from easier to harder examples aligns with our goal of improving model capabilities on more challenging reasoning tasks. Benchmark-specific performance breakdowns are provided in Appendix C. Our complete Bradley-Terry rating methodology is described in Appendix E.

### 5.4 ABLATION STUDIES ON THE VERIFIER

**Non-target Model Verification.** We investigate the impact of verification strategy in our SynthRL pipeline (Table 3). When using a non-target model (`Gemini-2.0-Flash-001` instead of

Table 3: Ablation study on different verifier configurations using 4K seed data.

| Verifier | Benchmark Accuracy (%) | | | | | |
|---|---|---|---|---|---|---|
| | MathVerse | MathVision | MathVista | WeMath | DynaMath | Avg. |
| $\mathcal{A}$-MMK12 | **52.5** | **29.0** | **74.0** | 70.5 | **60.0** | **57.2** |
| *w/ non-target verifier* | 51.2 | 28.2 | 71.2 | 70.2 | 57.9 | 55.7 |
| *w/ single-pass verifier* | 51.9 | 28.9 | 72.5 | **70.7** | 58.3 | 56.5 |
| *w/o verifier* | 49.6 | 28.9 | 71.1 | 70.5 | 58.1 | 55.8 |

Table 4: Ablation study on data strategies using 4K seed data.

| Strategy | #Data | Benchmark Accuracy (%) | | | | | |
|---|---|---|---|---|---|---|---|
| | | MathVerse | MathVision | MathVista | WeMath | DynaMath | Avg. |
| MMK12 | 4K | 50.3 | **29.8** | 73.7 | 70.2 | 58.2 | 56.5 |
| $\mathcal{R}$-MMK12 | 4K | 51.2 | 29.4 | 71.7 | 69.7 | 58.0 | 56.1 |
| $\mathcal{A}$-MMK12 | 4K+1.6K | **52.5** | 29.0 | **74.0** | **70.5** | **60.0** | **57.2** |

`Qwen2.5-VL-7B-Instruct`) as verifier, average accuracy drops from 57.2% to 55.7%. This demonstrates that effective verification requires alignment with the target model's capabilities to properly calibrate difficulty.

**Single-pass Verification and Unverified Synthesis.** We also explore simplified verification approaches. Single-pass verification uses the target model but performs only one verification per question rather than multiple Monte Carlo rollouts, achieving 56.5% average accuracy. Unverified synthesis, which removes verification entirely, yields 55.8% average accuracy.

These results confirm that verification aligned with the target model and using Monte Carlo rollouts contributes approximately 1.4% to overall performance gains, highlighting verification's essential role in SynthRL's effectiveness.

## 5.5 ABLATION STUDIES ON DATA STRATEGY

We examine different strategies for integrating synthesized data into training. Table 4 compares our augmentation approach $\mathcal{A}$-MMK12 with a replacement strategy $\mathcal{R}$-MMK12, where synthesized samples replace their corresponding seed samples while maintaining the same dataset size. Results show $\mathcal{A}$-MMK12 achieves the highest average accuracy at 57.2% across the five benchmarks, while $\mathcal{R}$-MMK12 underperforms even the original baseline (56.1% vs. 56.5%). This suggests synthesized questions provide maximum benefit when complementing rather than replacing the original distribution. The performance gap confirms SynthRL's improvements stem from both data scaling and the targeted difficulty enhancement of the training data.

## 6 CONCLUSION

We present SynthRL, an automated pipeline that improves VLM reasoning with RLVR by synthesizing more challenging training data. SynthRL follows a three-stage process: selecting seed questions based on difficulty, generating harder variants via a strong VLM while preserving answers, and verifying correctness and increased difficulty under a highly guaranteed mechanism. Applied to the MMK12 dataset, SynthRL produced over 3,380 verifiable, challenging questions from 8,072 seeds. Models trained on this data achieved significant accuracy gain across five out-of-domain visual math reasoning benchmarks, with larger improvements on the hardest samples, suggesting enhanced reasoning. SynthRL offers a scalable, data-centric method to boost VLM reasoning through automated, verifiable data synthesis.

ETHICS STATEMENT

This work builds on publicly available datasets and employs large vision-language models for controlled data synthesis. All synthesized data undergoes rigorous verification to ensure correctness and prevent the propagation of harmful or misleading content. No personally identifiable or sensitive data was used in any stage of this research.

REPRODUCIBILITY STATEMENT

To support reproducibility, we provide the full training and evaluation code, along with detailed run instructions, in the supplementary materials. These materials include all hyperparameter settings, implementation details, and usage guidelines required to replicate our experiments under the same conditions. The synthesized data generated through our pipeline will be released after publication to ensure accessibility and foster further research. Together, these resources will allow other researchers to faithfully reproduce and extend our results.

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

# APPENDIX

## TABLE OF CONTENTS

## A  LIMITATIONS

The current study robustly demonstrates SynthRL's efficacy using a specific large vision-language model as the synthesizer and explores data scaling up to 8K seed samples. However, a comprehensive investigation into the broader scalability continuum, potentially involving an even wider range of data volumes or a comparative analysis across varied synthesizer model architectures, was beyond the scope of available computational resources. Elucidating these aspects further could provide deeper insights into optimizing the trade-offs between synthesis cost and performance upper bound, and remains a compelling direction for subsequent work.

## B  REINFORCEMENT LEARNING WITH VERIFIABLE REWARDS ALGORITHM

Group Relative Policy Optimization (GRPO) (Shao et al., 2024), originally designed for mathematical reasoning in LLMs, can be effectively adapted to enhance visual reasoning capabilities in VLMs. We

use reinforcement learning to update our VLM, rewarding it based on a task-specific reward function $r_f$, where the subscript $f$ indicates the task.

For an input pair $(I, \mathbf{q})$ consisting of an image and text query from the training distribution $p_{\mathcal{D}}$, we employ a rule-based reward function $r_{f,q}$ that assigns $r_{f,q} = 1$ when the generated response $\mathbf{o}$ correctly answers the query (as determined by a verifiable parser) and $r_{f,q} = 0$ otherwise. This binary reward design helps prevent reward hacking during optimization.

The reference policy $\pi_{\theta_{\text{old}}}$ generates $n$ response rollouts for each input. The normalized advantage for the $i$-th rollout is calculated as:

$$A_i^{\text{norm}} = \frac{r_{f,q} - \text{mean}(\{r_{f,q}\}^n)}{\text{std}(\{r_{f,q}\}^n)},$$

where mean and std are calculated across the $n$ rollouts. Building upon PPO (Schulman et al., 2017), the GRPO objective function is formulated as:

$$\mathcal{J}_{\text{GRPO}}(\theta) = \mathbb{E}_{(I,\mathbf{q}) \sim p_{\mathcal{D}}, \mathbf{o} \sim \pi_{\theta_{\text{old}}}(\cdot | I, \mathbf{q})}$$
$$\left[ \frac{1}{n} \sum_{i=1}^{n} \min \left( s_i(\theta) A_i^{\text{norm}}, \text{clip}(s_i(\theta), 1 - \epsilon, 1 + \epsilon) A_i^{\text{norm}} \right) \right], \quad (4)$$

where $s_i(\theta) = \frac{\pi_\theta(\mathbf{o}_i | I, \mathbf{q})}{\pi_{\theta_{\text{old}}}(\mathbf{o}_i | I, \mathbf{q})}$ is the probability ratio between the new and old policies, and $\epsilon > 0$ defines the clipping range. Following recent practices in Meng et al. (2025) and Liu et al. (2025b), we do not apply any KL penalty to the reward.

## C   DETAILED BENCHMARK PERFORMANCE BY DIFFICULTY LEVEL

To complement the averaged difficulty analysis in Section 5.2, we present detailed performance results for each benchmark across easy, medium, and hard difficulty levels in Table 5. This breakdown shows how SynthRL's improvements vary across individual benchmarks at all three data scales.

Table 5: Performance comparison between MMK12 and $\mathcal{A}$-MMK12 models across benchmark difficulty levels. Accuracy (%) on easy, medium, and hard problem subsets for each benchmark.

| Method | MathVerse | | | MathVision | | | MathVista | | | WeMath | | | DynaMath | | |
|---|---|---|---|---|---|---|---|---|---|---|---|---|---|---|---|
| | Easy | Med. | Hard | Easy | Med. | Hard | Easy | Med. | Hard | Easy | Med. | Hard | Easy | Med. | Hard |
| *2K* | | | | | | | | | | | | | | | |
| MMK12 | 64.2 | 43.0 | 46.2 | 33.3 | 25.2 | 26.9 | 84.5 | 72.9 | 49.4 | 85.9 | 69.1 | 62.9 | 68.0 | 61.0 | 39.2 |
| $\mathcal{A}$-MMK12 | 62.1 | 41.3 | 46.8 | 35.1 | 26.2 | 26.9 | 86.9 | 72.1 | 55.3 | 87.5 | 67.3 | 60.7 | 68.4 | 62.4 | 38.5 |
| $\Delta$ | -2.1 | -1.7 | +0.6 | +1.8 | +1.0 | 0.0 | +2.4 | -0.8 | +5.9 | +1.6 | -1.8 | -2.2 | +0.4 | +1.4 | -0.7 |
| *4K* | | | | | | | | | | | | | | | |
| MMK12 | 61.4 | 43.7 | 46.0 | 34.8 | 28.2 | 26.4 | 86.9 | 75.3 | 54.4 | 86.4 | 68.4 | 66.3 | 70.1 | 61.4 | 39.1 |
| $\mathcal{A}$-MMK12 | 64.0 | 44.7 | 48.5 | 33.1 | 28.2 | 25.9 | 86.2 | 75.3 | 56.5 | 84.8 | 68.8 | 69.1 | 70.3 | 63.0 | 40.4 |
| $\Delta$ | +2.6 | +1.0 | +2.5 | -1.7 | 0.0 | -0.5 | -0.7 | 0.0 | +2.1 | -1.6 | +0.4 | +2.8 | +0.2 | +1.6 | +1.3 |
| *8K* | | | | | | | | | | | | | | | |
| MMK12 | 64.1 | 42.7 | 47.5 | 35.7 | 27.5 | 26.5 | 89.0 | 74.2 | 54.9 | 89.1 | 68.8 | 65.7 | 68.8 | 61.6 | 39.4 |
| $\mathcal{A}$-MMK12 | 66.0 | 46.7 | 48.3 | 35.6 | 26.5 | 26.2 | 86.6 | 75.3 | 57.0 | 86.4 | 71.0 | 70.8 | 69.6 | 63.4 | 39.6 |
| $\Delta$ | +1.9 | +4.0 | +0.8 | -0.1 | -1.0 | -0.3 | -2.4 | +1.1 | +2.1 | -2.7 | +2.2 | +5.1 | +0.8 | +1.8 | +0.2 |

# D    ADDITIONAL DATA ANALYSIS

To complement the 8K dataset analysis presented in Section 4.2, we present the characteristics of our 2K and 4K dataset variants.

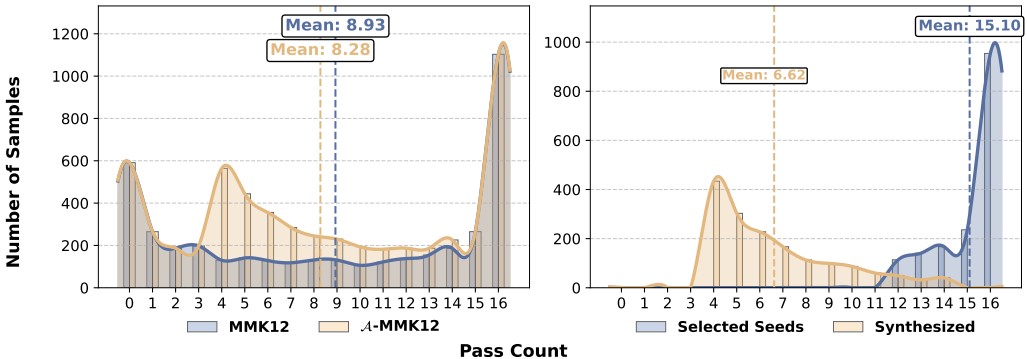

Figure 7: Pass rate distributions for the 4K dataset (4096 seed, 1612 synthesized). Consistent with the 8K dataset, synthesized questions show more balanced difficulty distributions compared to seed examples.

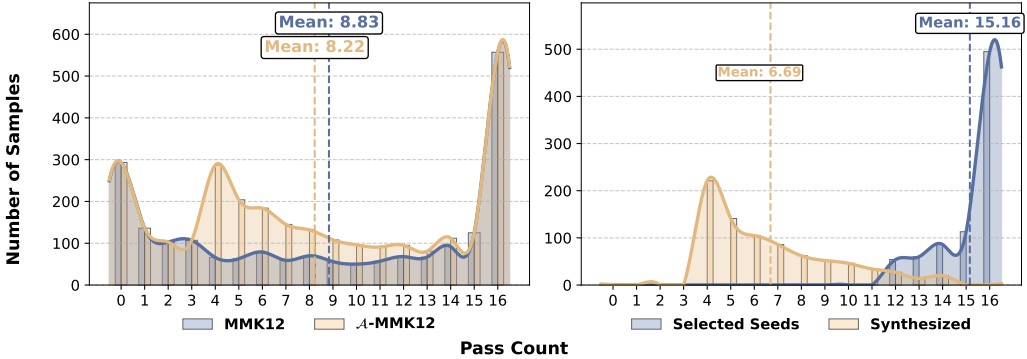

Figure 8: Pass rate distributions for the 2K dataset (2048 seed, 808 synthesized). Similar patterns are observed as in the 4K and 8K datasets, with synthesized questions displaying a more balanced distribution across difficulty levels.

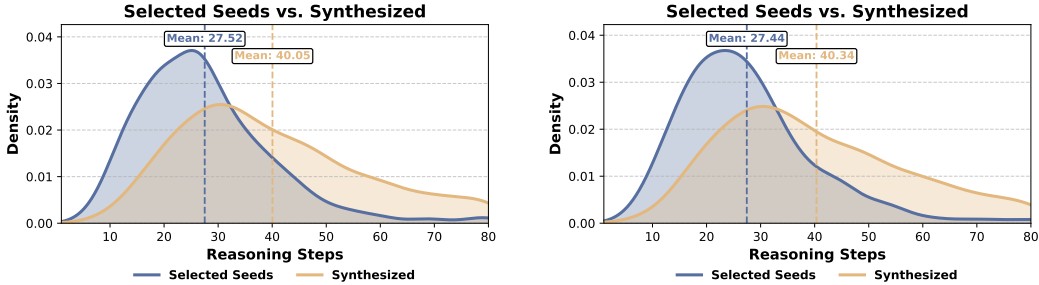

Figure 9: Distribution of reasoning steps between selected seed questions and synthesized questions for 4K and 2K datasets. In both cases, synthesized questions require more reasoning steps.

The 2K and 4K dataset variants exhibit similar characteristics to the 8K dataset. Figures 7 and 8 show that synthesized questions maintain a more balanced difficulty distribution compared to seed examples across all data sizes. Figure 9 confirms that the reasoning step patterns also remain consistent, with synthesized questions requiring more complex reasoning steps than their seed counterparts. These findings demonstrate that our synthesis approach produces consistent data quality regardless of the seed dataset size.

# E  BRADLEY-TERRY DIFFICULTY RATING METHODOLOGY

To systematically quantify the difficulty of data samples within our benchmarks, we employed the Bradley-Terry model (Bradley & Terry, 1952; Terry, 1952). This probabilistic model estimates latent difficulty parameters for items based on the outcomes of pairwise comparisons. These difficulty ratings enable the segmentation of each benchmark into easy, medium, and hard subsets.

The Bradley-Terry model posits that if $p_i$ is the positive real-valued difficulty parameter for sample $i$, the probability that sample $i$ is more difficult than sample $j$, denoted $P(i \succ j)$, is given by:

$$P(i \succ j) = \frac{p_i}{p_i + p_j} \tag{5}$$

By reparameterizing the difficulty parameters as $\theta_i = \log p_i$, the model can be expressed in a logistic form:

$$P(i \succ j) = \frac{e^{\theta_i}}{e^{\theta_i} + e^{\theta_j}} = \sigma(\theta_i - \theta_j) \tag{6}$$

where $\sigma(x) = 1/(1 + e^{-x})$ is the logistic sigmoid function. This formulation (Equation 6) connects the Bradley-Terry model to logistic regression frameworks, which are used for estimating the parameters $\theta_i$.

## E.1  PAIRWISE COMPARISON DATA COLLECTION

For each data sample, pairwise comparisons ("battles") were conducted against other samples from the same benchmark to establish relative difficulty. The specifics of this process were as follows:

- **MathVision, MathVista, and WeMath:** Each sample was compared against $k = 128$ randomly selected distinct samples from its respective dataset. This generated $128 \times N$ battle records for each dataset, where $N$ is the total number of samples in that dataset.
- **MathVerse:** This benchmark includes five versions for each problem instance, varying in visual-to-textual context ratio. Battles were performed exclusively on the "Text Lite" subset; each Text Lite sample was compared against 128 other Text Lite samples. The difficulty rating derived for a Text Lite sample was then assigned to its corresponding versions.
- **DynaMath:** This benchmark features 10 variants for each question. Battles were conducted using only "variant 1" of each question, with each such sample compared against 128 other variant 1 samples. The resulting difficulty rating was applied to its other 9 variants.

For every comparison pair, the difficulty evaluation was conducted using the `gemini-2.0-flash-001` model with a temperature setting of 0.6. To eliminate potential ordering bias, we randomized the presentation sequence of the two samples within each prompt. The specific prompt template used is:

---

**Difficulty Evaluation Prompt**

Compare math problems based on their difficulty. Consider reasoning steps, domain knowledge needed, and computational complexity in your assessment.
$< Image\ 1 >$
FIRST PROBLEM: $Problem\ 1$
$< Image\ 2 >$
SECOND PROBLEM: $Problem\ 2$
Which of these two math problems is more difficult?
Provide a brief explanation comparing their difficulty levels, then end with exactly one of: "WINNER: FIRST", "WINNER: SECOND", or "WINNER: TIE"

---

In this evaluation framework, the placeholders $Image\ 1$, $Problem\ 1$, $Image\ 2$, and $Problem\ 2$ were substituted with the visual content and textual descriptions of the mathematics problems being compared. For each target sample, we selected $k = 128$ opponent samples through random sampling without replacement from the pool of available unique opponents within the same dataset.

### E.2 Justification for the Number of Comparisons

Each sample underwent $k = 128$ pairwise comparisons. This number was chosen to support robust difficulty estimation, based on:

1. **Graph Connectivity:** The Bradley-Terry model requires a strongly connected comparison graph for unique Maximum Likelihood Estimates (MLEs) of its parameters $\theta_i$ (Ford Jr, 1957). We ensure that the comparison graph for each benchmark is connected, a necessary condition for the estimation of these parameters.

2. **Sufficient Data for Precise Parameter Estimation:** Beyond connectivity, $k = 128$ comparisons per sample provide substantial data for precise parameter estimation. Theoretical results for ranking from pairwise comparisons indicate that the maximum error of the estimated parameters (e.g., $\|\hat{\boldsymbol{\theta}} - \boldsymbol{\theta}^*\|_\infty$) can be bounded by terms proportional to $\sqrt{(\log N)/k_{\min}}$, where $N$ is the number of items and $k_{\min}$ is the minimum number of comparisons per item, provided $k_{\min} \gtrsim \log N$ (Hajek et al., 2014). For our largest benchmark, MathVision ($N = 3040$), our number of comparisons per sample $k = 128$ significantly exceeds $\log_2 N \approx 11.57$. This condition $k \gg \log N$ ensures the factor $\sqrt{(\log N)/k}$ is small, contributing to higher precision. This high number of comparisons per data sample provides a strong empirical basis for estimating the parameters, consistent with requirements for reliable parameter recovery in such models (Negahban et al., 2012). Consequently, this data volume supports stable and precise $\hat{\theta}_i$ estimates.

### E.3 Parameter Estimation and Elo Rating System

We estimated log-difficulty parameters $\hat{\theta}_i$ by fitting a logistic regression model to the pairwise comparison data. For each comparison between samples $a$ and $b$, we constructed a feature vector where the position for sample $a$ contains $+1$, sample $b$ contains $-1$, and all others are 0. Ties were handled by assigning 0.5 wins to each participant, and minimal L2 regularization was applied.

The estimated parameters were converted to an Elo-like rating scale:

$$\text{Elo}_i = \frac{S}{\ln(B)}\hat{\theta}_i + R_0 \tag{7}$$

where $S = 400$ is the Elo scale factor, $B = 10$ is the base (a 400-point difference representing 10:1 odds), and $R_0 = 1000$ is the baseline rating.

To assess stability and establish confidence intervals, we performed 100 rounds of bootstrapping with replacement on the comparison records. The final Elo rating for each sample is the median of its bootstrapped ratings, with 95% confidence intervals derived from the 2.5th and 97.5th percentiles. NaN values from any bootstrap sample were conservatively imputed with the minimum observed rating before calculating quantiles.

### E.4 Difficulty Level Categorization

Based on the final median Elo ratings, samples within each benchmark were categorized into three difficulty levels:

- **Hard:** Samples with an Elo rating $\geq 1050$.
- **Medium:** Samples with an Elo rating such that $950 < \text{Elo} < 1050$.
- **Easy:** Samples with an Elo rating $\leq 950$.

This categorization allows for a more granular analysis of model performance across varying degrees of problem complexity.

# F   TEMPLATES

> **Reasoning Template from EasyR1**
>
> **SYSTEM:** You are a helpful assistant.
> **USER:** You FIRST think about the reasoning process as an internal monologue and then provide the final answer.The reasoning process MUST BE enclosed within <think> </think> tags. The final answer MUST BE put in \boxed{}. {question}

# G   SUPPLEMENTARY IMPLEMENTATION DETAILS

This section provides the detailed hyperparameter configuration used in our implementation. Table 6 summarizes the configuration followed for all runs. We adjust training episodes based on dataset size to ensure convergence and obtain sufficient checkpoints for thorough evaluation.

Table 6: Summary of Hyperparameter Configurations

| Parameter | Configuration |
|---|---|
| **General Settings** | |
| Model Base | Qwen2.5-VL-7B-Instruct |
| Vision Encoder | Unfrozen |
| Max Prompt Length | 2048 tokens |
| Max Response Length | 2048 tokens |
| Max Image Pixels | 1,003,520 pixels |
| Min Image Pixels | 262,144 pixels |
| **Training Settings** | |
| Global Batch Size | 128 |
| Rollout Batch Size | 512 |
| Learning Rate | 1e-6 |
| Optimizer | AdamW |
| Grad Clip | 1.0 |
| Policy Loss Aggregation | `token-mean` |
| **RL Settings** | |
| Algorithm | GRPO (Appendix B) |
| KL Loss | False |
| KL Reward | False |
| Entropy Coefficient | 0.001 |
| $N$ Rollouts | 8 |
| Rollout Temperature | 1.0 |
| Rollout Top-P | 1.0 |
| **Training Episodes by Dataset Size** | |
| 2K variants | 100 steps |
| 4K variants | 144 episodes |
| 8K variants | 256 episodes |

## H  PSEUDOCODE FOR THE SYNTHRL PIPELINE

To better illustrate the SynthRL pipeline, Algorithm 1 presents the core verification procedure for synthesizing harder questions, while Algorithm 2 details the helper functions that enable the main procedure.

---

**Algorithm 1** SynthRL of a Single Harder Question

---

0: **Input:** Image $I$, original question $Q_{\text{ori}}$, original answer $A$,
  target policy $\pi_{\text{target}}$, synthesis VLM $\phi_{\text{synth}}$, judge model $M_{\text{judge}}$,
  solvability threshold $T_{\text{min}}$, min difficulty increase $\Delta_{\text{hard}}$,
  quality threshold $T_{\text{quality}}$, num synthesis attempts $N_{\text{attempts}}$, num rollouts $N$
0: **Output:** A single $Q_{\text{valid\_cand}}$ (validated harder question), or null
0: $c_{\text{ori}} \leftarrow$ CalculateRolloutPassCount($\pi_{\text{target}}, I, Q_{\text{ori}}, A, N$) {Establish baseline difficulty for $Q_{\text{ori}}$}
0: **for** $i = 1$ to $N_{\text{attempts}}$ **do**
0:  $Q_{\text{cand}} \leftarrow$ SynthesizeCandidateQuestion($\phi_{\text{synth}}, I, Q_{\text{ori}}$) {Generate candidate, $A$ is withheld from $\phi_{\text{synth}}$}
0:  $quality\_score \leftarrow$ AssessCandidateQuality($M_{\text{judge}}, I, Q_{\text{ori}}, Q_{\text{cand}}, A$) {Evaluate linguistic quality of $Q_{\text{cand}}$}
0:  **if** $quality\_score < T_{\text{quality}}$ **then**
0:   **continue** {Skip if below quality threshold}
0:  **end if**
0:  $c_{\text{cand}} \leftarrow$ CalculateRolloutPassCount($\pi_{\text{verifier}}, I, Q_{\text{cand}}, A, N$) {Evaluate difficulty of $Q_{\text{cand}}$} {Verify if $Q_{\text{cand}}$ is solvable and demonstrably harder}
0:  **if** $c_{\text{cand}} \geq T_{\text{min}}$ **and** $c_{\text{cand}} \leq c_{\text{ori}} - \Delta_{\text{hard}}$ **then**
0:   **return** $Q_{\text{cand}}$ {Return the first valid harder question found}
0:  **end if**
0: **end for**
0: **return** null {No suitable harder question found}
 =0

---

**Algorithm 2** Helper Functions for SynthRL

---

0: **function** CALCULATEROLLOUTPASSCOUNT($\pi_{\text{policy}}, I, Q, A, N_{\text{rollouts}}$)
0:  pass_count $\leftarrow 0$
0:  **for** $j = 1$ to $N_{\text{rollouts}}$ **do**
0:  $A_{\text{pred}} \sim \pi_{\text{policy}}(\cdot | I, Q)$ {Get predicted answer via stochastic forward pass}
0:  **if** $A_{\text{pred}}$ matches $A$ **then**
0:   pass_count $\leftarrow$ pass_count $+ 1$
0:  **end if**
0:  **end for**
0:  **return** pass_count {Return raw number of successful predictions}
0: **end function**

0: **function** SYNTHESIZECANDIDATEQUESTION($\phi_{\text{synth}}, I, Q_{\text{ori}}$)
0:  Prompt $\phi_{\text{synth}}$ with $(I, Q_{\text{ori}})$ to generate $Q_{\text{cand}}$
0:  {Original answer $A$ is not provided to $\phi_{\text{synth}}$}
0:  **return** $Q_{\text{cand}}$
0: **end function**

0: **function** ASSESSCANDIDATEQUALITY($M_{\text{judge}}, I, Q_{\text{ori}}, Q_{\text{cand}}, A$)
0:  Prompt $M_{\text{judge}}$ to rate quality of $Q_{\text{cand}}$
  (context: $I, Q_{\text{ori}}, A$)
0:  **return** quality score
0: **end function**
 =0

---

# I    CASE STUDY

To better illustrate the capabilities of our SynthRL approach, we provide four representative examples comparing the generated harder questions with their original counterparts.

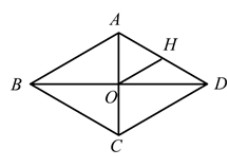

**Question:** As shown in the figure, in rhombus $ABCD$, diagonals $AC$ and $BD$ intersect at point $O$, $AC=6$, $BD=8$, and $H$ is the midpoint of side $AD$. Find the length of $OH$.
**Answer**: \frac{5}{2}

**Target Model Monte Carlo Rollout Pass**: 15 out of 16

ID: math_7715

- - - - - - - - - - - - - - - - - - - - - - - - - - - - - - - - - - - - - - - - - - - - -

**SynthRL**

**Question:** Consider a rhombus ABCD where the length of each side is 5 units. The area of the triangle formed by vertices A, B, and D is 12 square units. The diagonals AC and BD intersect at point O. Let H be a point located on the side AD such that the segment OH divides triangle AOD into two triangles of equal area. Determine the length of the segment OH.
**Answer**: \frac{5}{2}

**Target Model Monte Carlo Rollout Pass**: 4 out of 16

Figure 10: Comparison of SynthRL generated harder question and original question, case 1.

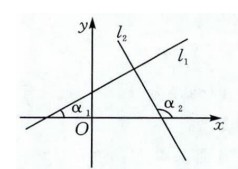

**Question:** As shown in the figure, the inclination angle of line $l_{2}$ is $\alpha_{2}=120^{\circ}$, and the inclination angle of line $l_{1}$ is $\alpha_{1}$. Given that $l_{1} \perp l_{2}$, what is the slope of line $l_{1}$?
**Answer**: /dfac{\sqrt{3}}{3}

**Target Model Monte Carlo Rollout Pass**: 15 out of 16
ID: math_5028

- - - - - - - - - - - - - - - - - - - - - - - - - - - - - - - - - - - - - - - - - - - - -

**SynthRL**

**Question:** Let $\theta$ be the unique angle in the interval $(0, \pi/2)$ such that the area of the triangle with vertices at the origin, the point representing $e^{i\theta}$, and the point representing $e^{i2\theta}$ in the complex plane is $\frac{\sqrt{3}}{4}$.
Let $\mathbf{p}$ be the vector from the origin to the point in the complex plane representing $e^{i\theta}$.
Let $\mathbf{s}$ be the vector obtained by rotating $\mathbf{p}$ counterclockwise by an angle of $\frac{\pi}{3}$ radians.
Line $l_2$ is parallel to the vector $\mathbf{s}$.
The line $l_1$ is oriented such that the directed angle from $l_2$ to $l_1$ is $\frac{\pi}{2}$ radians.
Determine the slope of line $l_1$.
**Answer**: /dfac{\sqrt{3}}{3}

**Target Model Monte Carlo Rollout Pass**: 5 out of 16

Figure 11: Comparison of SynthRL generated harder question and original question, case 2.

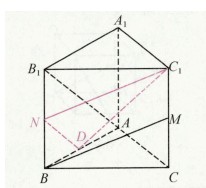

**Question:** As shown in the figure, $BC \perp AE$, with the foot of the perpendicular at $C$. A line through $C$ is drawn parallel to $AB$, and if $\angle ECD = 43^\circ$, then $\angle B$ is ____ degrees?

**Answer:** 47

**Target Model Monte Carlo Rollout Pass:** 12 out of 16

**ID: math_7140**

---

SynthRL

**Question:** As shown in the figure, a line is drawn through $C$ parallel to $AB$, intersecting ray CE at D, where E is a point on the extension of AC. If $\angle ABC$ and $\angle CAB$ are complementary, and the difference between $\angle ABC$ and $\angle ECD$ is $4{}^\circ$, then what is the measure of $\angle ABC$ in degrees?

**Answer:** 47

**Target Model Monte Carlo Rollout Pass:** 6 out of 16

Figure 12: Comparison of SynthRL generated harder question and original question, case 3.

**Question:** As shown in the figure, in the regular triangular prism $ABC-A_{1}B_{1}C_{1}$, all edges are of equal length. Point $M$ is the midpoint of the side edge $CC_{1}$. What is the angle formed by the skew lines $AB_{1}$ and $BM$ in degrees?

**Answer:** 90

**Target Model Monte Carlo Rollout Pass:** 13 out of 16

**ID: math_7715**

---

SynthRL

**Question:** In a regular triangular prism $ABC-A_{1}B_{1}C_{1}$ where all edges are of equal length, let $M$ be the midpoint of the side edge $CC_{1}$. Let $L$ be the locus of points $P$ in space such that the vector $\vec{AP}$ is orthogonal to the vector $\vec{BM}$, and the vector $\vec{BP}$ is orthogonal to the vector $\vec{A B_{1}}$. Determine the measure of the angle between the line $AB_{1}$ and the line $L$ in degrees.

**Answer:** 90

**Target Model Monte Carlo Rollout Pass:** 5 out of 16

Figure 13: Comparison of SynthRL generated harder question and original question, case 4.

## J BROADER IMPACT

SynthRL addresses a critical challenge in developing visual reasoning models by automating the creation of verified, challenging training examples that would otherwise require extensive human annotation. By generating high-quality, guaranteed-correct data for reinforcement learning, our approach significantly reduces the time-consuming and costly human labeling process typically required for RL training data. This automation enables researchers to scale up training datasets with diverse, difficulty-controlled examples, potentially democratizing access to robust visual reasoning capabilities across research communities with varying resource constraints.

## K LICENSES

We use standard licenses from the community. We include the following licenses for the codes, datasets and models we used in this paper.

Datasets & Benchmarks:

- MMK12 (Meng et al., 2025): Apache License 2.0
- MathVerse (Zhang et al., 2024a): MIT
- MathVision (Wang et al., 2024a): MIT
- MathVista (Lu et al., 2023): Creative Commons Attribution Share Alike 4.0 International
- WeMath (Qiao et al., 2024): CC BY-NC 4.0
- DynaMath (Zou et al., 2024): Creative Commons Attribution Share Alike 4.0 International

Codes:

- verl (Sheng et al., 2024): Apache License 2.0
- EasyR1 (Zheng et al., 2025): Apache License 2.0

Models:

- Qwen2.5-VL-7B-Instruct (Bai et al., 2025): Apache License 2.0
- Gemini API (Gemini Team, 2023): Gemini API Additional Terms of Service

## L USE OF LLMs

Large language models (LLMs) were employed solely as auxiliary tools to support writing, debugging, and data synthesis. Specifically, LLMs were used to improve sentence clarity, grammar, and overall flow of the manuscript, as well as to provide occasional assistance in resolving minor implementation issues (e.g., detecting syntax errors or suggesting refactoring options). In addition, LLMs were utilized in controlled settings to aid in the synthesis of training data, particularly in generating candidate questions later subjected to rigorous verification. Importantly, all core research ideas, methodological designs, experimental frameworks, codebase implementations, and scientific analyses were entirely conceived, developed, and conducted by the authors. No component of the intellectual contribution, experimental reasoning, or scientific conclusions was produced by an LLM.

