# OpenReview forum: "SynthRL: Scaling Visual Reasoning with Verifiable Data Synthesis"
_ICLR.cc/2026/Conference — ICLR 2026 Conference Withdrawn Submission_

### Official Review · Reviewer_cPJ9 · 2025-10-29

**Soundness:** 3
**Presentation:** 3
**Contribution:** 2
**Rating:** 4
**Confidence:** 4

**Summary:**

The paper proposes a pipeline to synthesize additional verifiable data from existing seed data. Specifically, it generates 3.3K synthetic samples from 8K seed samples through a three-stage process: seed question selection, question difficulty enhancement, and verification.

**Strengths:**

- Addresses an important problem in reinforcement learning (RL) data synthesis.
- Provides a comprehensive ablation study in Table 3, particularly for the non-target verifier (though further explanation of these results would strengthen the paper).

**Weaknesses:**

- The scalability of the approach is unclear—specifically, how many new data points can be generated per seed sample.
- The improvement reported (<2 points) is relatively marginal, especially on reasoning-intensive math and visual datasets.

**Questions:**

- [Suggestion] In Line 224, the term “correctness criterion” may be misleading, as the criterion does not guarantee correctness. The added transformations could introduce ambiguity that models occasionally interpret correctly.
- Could the authors provide more qualitative examples illustrating how easy questions are transformed into harder ones? Figure 4 shows changes in pass rates but does not clarify the transformation process.
- In Figure 5, how is the number of reasoning steps computed?
- Generating only 3.3K samples from 8K seed data seems modest. What is the main bottleneck? For instance, does the need for high $T_\text{min}$ and $\Delta_\text{hard}$ limit scalability?

---

### Official Review · Reviewer_YJRg · 2025-10-30

**Soundness:** 2
**Presentation:** 3
**Contribution:** 2
**Rating:** 4
**Confidence:** 4

**Summary:**

This paper addresses the data-scaling bottleneck in RLVR for vision–language reasoning. It introduces SynthRL, a three-stage pipeline. First, the target model’s Monte Carlo pass counts are used to score seed items to identify easy but correct questions. Second, a stronger VLM synthesizes harder variants that preserve the original answer. Third, multi-sample verification aligned with the target model enforces both correctness and increased difficulty. The method expands MMK12 with 3,380 validated items (A-MMK12) and delivers consistent gains across five visual math benchmarks, with the largest improvements on medium and hard subsets. Ablations demonstrate that verifier alignment and multi-sample checks are crucial, and that augmenting seeds rather than replacing them is more effective. Overall, SynthRL provides a practical, verifiable way to scale RLVR training data for small yet reliable improvements in reasoning.

**Strengths:**

1.The paper is logically clear and easy to follow.
2.Verification with multi-sample checks preserves answers and increases difficulty.
3.Experiments show consistent improvements in accuracy across five visual math benchmarks, with larger effects on medium and hard subsets.

**Weaknesses:**

1.The distinction from teacher-driven generative distillation is unclear. Prior work also relies on a larger model to rewrite inputs [1][2].

2.The answer consistency verification relies on the model-dependent solvability check rather than ground-truth guarantees. Prompt guidance does not enforce the same answer and shifted answers can still pass.

3.A train-test overlap auditing is suggested. The authors do not report explicit deduplication or coverage checks between the training data and the evaluation sets.

4.There is single-model dependence for both synthesis and evaluation without human oversight. Generation uses Gemini-2.5-Flash-Preview-04-17, and scoring and difficulty checks rely on Gemini-2.0-Flash-001 as the sole judge, with no multi-judge validation or manual audits reported.

5.Overall accuracy improvements are marginal and may be sensitive to randomness in training and synthesis. Reported improvements are around 0.2–1.0% across scales. Results rely on a single best checkpoint with no multi-run variance. The easy subset shows a noticeable decline.

6.SynthRL is not compared against alternative multimodal data augmentation methods. Experiments only contrast internal variants with the MMK12 seed baseline.

[1] Xu C, Sun Q, Zheng K, et al. WizardLM: Empowering large pre-trained language models to follow complex instructions[C]//The Twelfth International Conference on Learning Representations. 2024.

[2] Luo R, Zhang H, Chen L, et al. Mmevol: Empowering multimodal large language models with evol-instruct[J]. arXiv preprint arXiv:2409.05840, 2024.

**Questions:**

1.An ablation of the solvability and difficulty criteria is suggested. The authors fix N=16, T_min=4, a difficulty margin of 2, and an easy-seed filter of 12/16 without sensitivity analyses.

2.Cross-domain generalization remains unclear given training on MMK12 and evaluation on visual math. Does SynthRL generalize to other multimodal reasoning domains in a small-scale transfer setting?

---

### Official Review · Reviewer_NiC7 · 2025-10-30

**Soundness:** 3
**Presentation:** 3
**Contribution:** 3
**Rating:** 4
**Confidence:** 4

**Summary:**

Interesting discovery on RL dataset synthesis scaling-up on math domain. The paper shows the insights of the importance of the lack of difficulty balancing, which brings the sparse reward during RL training. By the proposed three-step approach, the paper shows better performances on math benchmarks than several other competitors with similar parameter size. The performance shows that the proposed method's effectiveness on data quality and effectiveness on future multimodal reasoning tasks.

**Strengths:**

1. The paper is well-written and the clearness and effectiveness of the method need to be praised.

2. The proposed approach is straightforward and provides reasonable performance to prove the insights that provided.

3. The results show the potential of scaling-up current math VQA dataset for larger-scale training to improve the generalization capability.

**Weaknesses:**

1. The scaling-up capability of the dataset on math domain may be limited. We can see that in table 1, with the increasing number of selected real data from 2K, 4K to 8K, the momentum of the increasing performance saturated quickly on MathVision, DynaMath, MathVista. For MathVIsion, the accuracy actually decreases when scaling-up from 2k to 4k. Similar saturation pattern also found in other 3 benchmarks.

2. There are 8k seed VQA questions selected but only 3.3k met the requirement of proposed approach. This limits the valuable dataset, and limits the scaling-up volume. A better design of the pipeline may be introduced. Moreover, the correctness of generated reasoning step is not equal to the correctness of the answers. It is the implicit assumption of the generated data.

3. The proposed method only focus on synthesizing multi-model math datasets and benchmarks. If the model is expected for multiple tasks including coding, visual perception, distance estimation, etc. How does this method scaled-up to other un-verifiable domains? The contribution maybe limited.

4. To use the same synthesized dataset, but by SFT without RL training may be a good ablation competitors to prove the specific contribution of the proposed method to RL scaling-up. We did not see it yet.

5. Together with 3, the increasing of using synthesized dataset by RL only increases a very marginal portions of the performance for all benchmarks (table 1, MMK12 rows compared to A-MMK12 rows). It limits the novelty and effectiveness contribution.

**Questions:**

1. It will be helpful to show a better justification of data scaling-up curve (and also a longer curve to >8K, because other methods seems included 10K-25K total) through proposed method, compared to other methods.

2. A detailed discussion will be helpful to explain why the proposed method works better, and why for some benchmark is does not work so well (MathVision, etc.)

3. A justification on domain generalization of the proposed method may be helpful to better frame the contribution of this paper.

4. A better justification of the dataset synthesizing for RL training rather SFT may be helpful for a better justification of the paper's contribution.

---

### Official Review · Reviewer_sgsW · 2025-10-31

**Soundness:** 3
**Presentation:** 3
**Contribution:** 2
**Rating:** 4
**Confidence:** 3

**Summary:**

### Summary

This paper proposes a **scalable and automated framework** for **reinforcement learning with verifiable rewards (RLVR)** in vision-language models (VLMs).
The core idea is to **synthesize new, more challenging yet verifiably correct training samples** to expand reasoning-oriented RL datasets without requiring human annotation.

The proposed **SynthRL** pipeline consists of three main stages:

1. **Difficulty-Based Seed Selection** –
   Selects "too easy" questions from the seed dataset (**MMK12**) by using **Monte Carlo rollout pass counts** to identify high-success-rate samples that can be made more challenging.

2. **Targeted Synthesis** –
   Utilizes a strong VLM (**Gemini-2.5-Flash**) to generate **harder variants** of the selected questions while preserving their **ground-truth answers**, encouraging deeper reasoning behavior.

3. **Guaranteed Verification** –
   Employs the **target model itself (Qwen2.5-VL-7B)** as a verifier to ensure that synthesized samples are both
   (a) **correct** (same final answer) and
   (b) **more difficult** (lower pass rate).
   Only samples meeting both criteria are retained, ensuring high data quality and measurable difficulty increase.

Applied to **8K seed samples** from the MMK12 dataset, SynthRL produces **3.3K verified harder samples**, forming the augmented dataset **A-MMK12** (11.4K total).
The synthesized data exhibit **balanced difficulty distribution** and **longer reasoning chains**, suggesting more complex and informative learning signals.

Models are trained with the **GRPO algorithm** under the RLVR setting and evaluated on **five out-of-domain visual reasoning benchmarks**: *MathVerse, MathVision, MathVista, WeMath,* and *DynaMath.*
**SynthRL-7B** consistently outperforms all baselines across these benchmarks (average **+1.0% improvement** at 8K scale).
The augmented dataset improves **generalization and training stability**, with larger benefits observed as the dataset scale increases.

**Strengths:**

1. **Consistent Improvements Across Multiple Benchmarks**
   Fine-tuning under the **SynthRL** framework consistently improves performance across **five diverse visual reasoning benchmarks** — *MathVerse, MathVision, MathVista, WeMath,* and *DynaMath.*
   Notably, **SynthRL-7B** achieves the **largest gains on medium and hard difficulty subsets**, demonstrating that the synthesized data effectively enhances **deep reasoning capabilities** rather than superficial pattern recognition.

2. **Effective Use of the Target Model as Verifier**
   Employing the **target model itself as the verifier** proves to be a highly effective design choice.
   This approach ensures that the synthesized data **aligns with the model’s own perception and reasoning distribution**, resulting in **more stable, consistent, and scalable RL improvements**.

**Weaknesses:**

1. **Limited Dataset and Domain Generalization**
   The experiments are conducted **only on the MMK12 dataset** (visual math reasoning for K–12), which limits the generalizability of the conclusions.
   It remains unclear whether the proposed **synthesis and verification pipeline** would perform equally well on **non-mathematical visual reasoning tasks**, such as diagram understanding, commonsense VQA, or scientific figure reasoning.

2. **Lack of Qualitative Examples and Failure Analysis**
   The paper reports strong quantitative gains but provides **few concrete examples** of synthesized samples, reasoning chains, or failure cases.
   For a **data-generation framework**, the inclusion of **visualizations or qualitative comparisons** between original and synthesized items would be valuable to verify that the increased difficulty is **semantically meaningful**, rather than merely surface-level or lexical.

**Questions:**

1. Using the same model family (Qwen2.5-VL) for both training and verification might reinforce existing biases or reasoning patterns.  Did the authors observe any **bias drift or overfitting** to the verifier’s implicit priors over multiple synthesis–verification cycles?

2. Since SynthRL improves data quality via synthesis and verification, how does it compare to simply **SFT** on the same augmented data, without reinforcement learning?

---

### Note · Authors · 2026-01-05

I have read and agree with the venue's withdrawal policy on behalf of myself and my co-authors.